# Non-Alcoholic Fatty Liver Disease, and the Underlying Altered Fatty Acid Metabolism, Reveals Brain Hypoperfusion and Contributes to the Cognitive Decline in APP/PS1 Mice

**DOI:** 10.3390/metabo9050104

**Published:** 2019-05-25

**Authors:** Anthony Pinçon, Olivia De Montgolfier, Nilay Akkoyunlu, Caroline Daneault, Philippe Pouliot, Louis Villeneuve, Frédéric Lesage, Bernard I. Levy, Nathalie Thorin-Trescases, Éric Thorin, Matthieu Ruiz

**Affiliations:** 1Department of Pharmacology and Physiology, Faculty of Medicine, Université de Montréal, Montreal, QC H3T 1J4, Canada; anthony.pincon@live.fr (A.P.); olivia.demontgolfier@gmail.com (O.D.M.); 2Research Center, Montreal Heart Institute, University of Montreal, Montreal, QC H1T 1C8, Canada; nilay.akk@gmail.com (N.A.); caroline.daneault@icm-mhi.org (C.D.); louis.villeneuve@icm-mhi.org (L.V.); nathalie.trescases@icm-mhi.org (N.T.-T.); 3Department of Electrical Engineering, Ecole Polytechnique de Montréal, Montreal, QC H3T 1J4, Canada; ph.pouliot@gmail.com (P.P.); frederic.lesage@polymtl.ca (F.L.); 4Institut des Vaisseaux et du Sang, Hôpital Lariboisière, 75010 Paris, France; bernard.levy@inserm.fr; 5Department of Surgery, Faculty of Medicine, Université de Montréal, Montreal, QC H3T 1J4, Canada; eric.thorin@umontreal.ca; 6Department of Medecine, Faculty of Medicine, Université de Montréal, Montreal, QC H3T 1J4, Canada

**Keywords:** NAFLD, inflammation, polyunsaturated fatty acids, senescence, brain hypoperfusion, untargeted and targeted lipidomics, amyloïd Beta, liver-to-brain axis, Alzheimer disease mouse model

## Abstract

Non-alcoholic fatty liver disease (NAFLD), the leading cause of chronic liver disease, is associated with cognitive decline in middle-aged adults, but the mechanisms underlying this association are not clear. We hypothesized that NAFLD would unveil the appearance of brain hypoperfusion in association with altered plasma and brain lipid metabolism. To test our hypothesis, amyloid precursor protein/presenilin-1 (APP/PS1) transgenic mice were fed a standard diet or a high-fat, cholesterol and cholate diet, inducing NAFLD without obesity and hyperglycemia. The diet-induced NAFLD disturbed monounsaturated and polyunsaturated fatty acid (MUFAs, PUFAs) metabolism in the plasma, liver, and brain, and particularly reduced n-3 PUFAs levels. These alterations in lipid homeostasis were associated in the brain with an increased expression of *Tnfα*, *Cox2*, *p21*, and *Nox2*, reminiscent of brain inflammation, senescence, and oxidative stress. In addition, compared to wild-type (WT) mice, while brain perfusion was similar in APP/PS1 mice fed with a chow diet, NAFLD in APP/PS1 mice reveals cerebral hypoperfusion and furthered cognitive decline. NAFLD reduced plasma β_40_- and β_42_-amyloid levels and altered hepatic but not brain expression of genes involved in β-amyloid peptide production and clearance. Altogether, our results suggest that in a mouse model of Alzheimer disease (AD) diet-induced NAFLD contributes to the development and progression of brain abnormalities through unbalanced brain MUFAs and PUFAs metabolism and cerebral hypoperfusion, irrespective of brain amyloid pathology that may ultimately contribute to the pathogenesis of AD.

## 1. Introduction

The brain is one of the most lipid-rich organs [1]. Cholesterol and fatty acids play a structural role in cell membranes and are involved in various biological functions. Brain cholesterol is essential for synaptogenesis [2], axonal growth [3], neurotransmitter release [4], and neuronal plasticity [5], while n-3 and n-6 polyunsaturated fatty acids (PUFAs) and their metabolites modulate inflammation [6,7,8], vasoreactivity [9], neurogenesis [10], brain glucose uptake [11,12], and cognition [6,7,13].

Clinical studies have shown that fatty acid and cholesterol levels are altered in the brains of Alzheimer’s disease (AD) patients [14,15,16,17]. Moreover, membrane cholesterol levels regulate the production, distribution and insertion of the β-amyloid peptide (Aβ), a pathological hallmark of AD [18,19], while n-3 PUFAs promote Aβ clearance from the brain [20], decrease brain inflammation and modulate the immune response to Aβ peptide [21]. While less investigated, a similar reasoning may be applied to monounsaturated fatty acids (MUFAs) and especially with the ability of palmitoleic acid (C16:1n7) and oleic acid (C18:1n9) to reduce Aβ levels [22]. Moreover, amyloidosis is improved with oleic acid supplementation [23]. In addition, lower MUFAs have been associated with worse cognitive performance [17]. However, data do not support unanimously the hypothesis that brain fatty acid and cholesterol abundance is a sensitizing factor for patients at risk of AD.

Dietary and nutrient pattern modulate cerebral lipid composition [24] and brain function. Low intake of n-3 PUFAs and high intake of cholesterol have been reported to be associated with cognitive decline in a multiethnic elderly population [25]. In contrast, a regular intake of n-3 PUFAs decreases the age-dependent cognitive decline and slows down the progression of AD in mice and humans [26,27]. Moreover, n-3 PUFA supplementation improves blood brain perfusion in patients with mild cognitive impairments [28], and plasma n-3 PUFA levels positively correlate with cerebral blood flow in humans [29]. Interestingly, clinical studies have demonstrated that non-alcoholic fatty liver disease (NAFLD), the leading cause of chronic liver disease [30], is associated with impaired cognitive function [31] and that liver n-3 PUFA content strongly correlates with the cognitive status in patients with AD [32], suggesting the existence of a liver-brain axis. However, the mechanisms underlying these interactions are unknown. In this study, we hypothesized that NAFLD could reveal the appearance of brain disturbances especially cerebral hypoperfusion in association with dysregulation of plasma and brain lipid metabolism in the APP/PS1 mouse model of AD [33].

## 2. Results

Since experimental data have shown that cerebral perfusion is decreased in patients with AD [34], and based on our observations of unchanged contralateral (Appendix A), ipsilateral (Appendix A) and global brain perfusion (Appendix A) in 6-month old APP/PS1 mice compared to wild-type (WT), fed with a standard diet, we aimed to evaluate whether lower brain perfusion in APP/PS1 mice may be revealed as a results of NAFLD development. Therefore, we fed the APP/PS1 mice with either a low-fat diet or a Paigen cholesterol-enriched diet (CHOL) to characterize the development of NAFLD, the underlying lipidomic changes, and the resulting consequences in the brain, our primary outcomes being brain perfusion and potential cognitive alterations.

### 2.1. CHOL Diet Induces the Phenotype of NAFLD without Obesity in APP/PS1 Mice

Weight gain in APP/PS1 mice fed a CHOL diet for 18 weeks was lower than that measured in mice fed a STD diet (delta weight: 3.6 ± 0.2 versus 4.5 ± 0.3 g, p < 0.05; Figure 1A,B) despite similar food intakes between groups (Figure 1C,D). Fasted plasma cholesterol (Figure 1E) and triglycerides (Figure 1F) levels were 65% (p < 0.001) and 157% (p < 0.01) higher in mice fed a CHOL diet than mice fed a STD diet, respectively. Fasted plasma glucose levels were similar between groups (Figure 1G). Liver cholesterol (Figure 1H) and triglycerides (Figure 1I) levels were 209% (p < 0.001) and 30% (p < 0.01) higher in mice fed a CHOL diet than mice fed a STD diet, respectively. Liver weight in mice fed a CHOL diet was 117% (p < 0.001; Figure 1J) higher compared to the liver of mice fed a STD diet. Altogether, these data show that the CHOL diet induced hallmarks of NAFLD in the absence of the development of obesity in APP/PS1 mice. To complete the phenotyping of NAFLD-related changes in liver, we observed changes in transcript levels of inflammatory and oxidative stress markers. Compared to mice fed a STD diet, *Tnfα* mRNA expression increased by 258% in livers from mice under CHOL diet while there was no change in *Il1β* mRNA expression (Figure 2A).

Similarly, in mice fed with a CHOL diet, we noticed 168% and 226% elevation in hepatic heme oxygenase-1 (*Ho-1*) and *Nox2* mRNA expression, respectively (Figure 2B) and a lower mRNA expression of *Sod1* (−32%), *Sod2* (−40%) and *Nox4* (−50%) (Figure 2B). Altogether, these results support that the CHOL diet induced the increase of a surrogate marker of inflammation in the liver and impaired hepatic oxidative stress balance, which are both hallmarks of NAFLD [35].

### 2.2. NAFLD is Associated with n-3 PUFAs Deficiency in the Triglyceride Fraction of the Plasma and the Liver

We then measured fatty acids profile in the triglycerides fraction of the liver (Figure 3A) and the plasma (Figure 3B). CHOL mice displayed many changes in most of the fatty acid (FA) measured and especially lower amounts of n-3 PUFAs in the liver compared to mice fed a STD diet, such as docosahexaenoic acid (DHA, C22:6n3) (p < 0.001), eicosapentaenoic acid (EPA, C20:5n3) (p < 0.001), and docosapentaenoic acid (DPA, C22:5n3) (p < 0.001) contents (Figure 3A). However, the levels of MUFAs increased and particularly C16:1n7 and C18:1n9 (p < 0.001).

While changes in plasma were less generalized, CHOL mice also exhibited lower levels of plasma n-3 PUFAs comprising the reduction of DHA (p < 0.05), EPA (p < 0.05) and DPA (p < 0.01) contents (Figure 3B). As in the liver, we observed an increased in MUFAs levels (p < 0.01) (Figure 3B). Similar results were observed in the phospholipid fraction of the liver (Appendix A) while changes were less obvious in the phospholipid fraction of the plasma (Appendix A). Altogether, these results show that NAFLD is associated with lower n-3 PUFAs in the plasma and the liver while MUFAs were enhanced.

### 2.3. NAFLD is Associated with a Dysregulated Brain Lipid Metabolism Particularly MUFAs and PUFAs Deficiency

To evaluate the impact of NAFLD on cerebral lipid metabolism, whole brain fatty acid profiles were performed. Interestingly, n-3 PUFAs deficiency was similar than that observed in plasma as regards of EPA (p < 0.05) and DPA (p < 0.05) in the brain of mice with NAFLD compared to control mice (Figure 3C). However, while increased in plasma, MUFAs comprising palmitoleic acid (C16:1n7; p < 0.01), oleic acid (C18:1n9; p < 0.05), vaccenic acid (C18:1n7; p < 0.05) decreased in the brain. A similar observation was made to some extent for n-6 PUFAs such as linoleic acid (C18:2n6; p < 0.05) and dihomo-γ-linolenic acid (C20:3n6; p < 0.05) (Figure 3C) which decreased only in the brain. The latter observations may suggest disturbed MUFAs and PUFAs metabolism in the brain and opposite changes in MUFA levels in the brain support a disconnection with the changes observed in the liver and the circulation.

To further characterize the brain lipidome, we used untargeted lipidomic analysis. According to objective threshold criteria’s, namely a false discovery rate (FDR) of 37.5 (corresponding to a p-value of 0.015) and a fold change (FC) of 1.2, we selected 70 MS signals significantly different between groups, among which 39 lipids were identified using MSMS analysis. Consistent with the above findings, most of the changes (down) were related to n-3 and n-6 PUFAs but included also changes (lower) in MUFAs (Appendix A). We also assessed the levels of brain cholesterol, of its precursors (desmosterol and lathosterol) and of its main metabolite (24S-hydroxychlesterol) (Appendix A): desmosterol levels were 0.76 fold lower (p < 0.001) and 24S-hydroxycholesterol contents tended (p = 0.095) to be slightly (5%) higher in the brain of mice with NAFLD compared to control mice (Appendix A), with no change in cholesterol and lathosterol levels (Appendix A). Altogether, these results show that NAFLD is associated with global cerebral PUFAs disturbances.

### 2.4. NAFLD Induces a Gene Expression Remodeling Reminiscent of Brain Inflammation, Cellular Senescence, and Oxidative Stress

We then sought to characterize the molecular impact of a dysregulated cerebral lipid homeostasis on inflammation, senescence, and oxidative stress in the brain. First, we assessed mRNA expression of cytokines in the brain of APP/PS1 mice: *Tnfα* mRNA expression was 38% higher (p < 0.05) in the brain of mice with NAFLD compared to control mice (Figure 4A), with no changes in *Il6*, *Il1β* and *Tgfβ* expressions (Figure 4A). Second, to investigate any associations between *Tnfα* overexpression and actors involved in neuro-inflammation, we evaluated transcript levels of astrocyte (*Gfap*), macrophages (*Cd68*) and adhesion cells (*Vcam*, *Icam*) markers (Figure 4B). There was no difference among groups. In contrast, we observed 62% and 52% higher mRNA expression of *Cox2* and *p21,* respectively, in mice with NAFLD compared to control mice (Figure 4B), suggesting the occurrence of both inflammation and senescence in relation to the dysregulated n-6 and n-3 PUFA metabolism. Finally, we quantified the relative expression of gene involved in the regulation of oxidative stress balance: we observed 35% (p < 0.05) higher *Nox-2* mRNA expression (Figure 4C) with no change in *Ho-1*, *Sod1* or *Sod2* mRNA expressions between groups (Figure 4C). Altogether, these results suggest that NAFLD is likely associated with inflammation, senescence, and some oxidative stress in the brain.

### 2.5. NAFLD is Associated with Dysregulated Hepatic and Plasma Aβ Metabolism

To investigate whether NAFLD may influence amyloid pathology, we first quantified plasma Aβ_40_ and Aβ_42_ levels by enzyme-linked immunosorbent assay (ELISA) (Figure 5A,B): plasma Aβ_40_ levels were 28% lower in mice with NAFLD than in control mice (Figure 5A). Unlike in the control group, we did not detect Aβ_42_ in the plasma of mice with NAFLD (Figure 5B), suggesting that NAFLD decreases not only plasma Aβ_40_ but also plasma Aβ_42_ peptide contents. Then, we assessed hepatic mRNA expression of genes involved in the production (*Bace-1*), catabolism (*Ide*, *Npe*) and clearance (*Lrp1*, *Ldlr*, *Lsr*) of Aβ peptides (Figure 5C): we observed 58% (p < 0.001), 71% (p < 0.001) and 45% (p < 0.05) reduction in the expression of *Bace-1*, *Ide*, and *Npe,* respectively, in the liver of mice fed a CHOL diet compared to mice fed a STD diet (Figure 5C). NAFLD also reduced by 78% (p < 0.001) and 62% (p < 0.05) hepatic mRNA expression of *Ldlr* and *Lrp1,* respectively, with no change in *Lsr* expression (Figure 5C). Altogether these data suggest that NAFLD was associated with a reduced production, catabolism and clearance of hepatic and plasma Aβ peptide.

Then, we sought to characterize the impact of NAFLD on amyloid pathology in the brain: neither the soluble Aβ_40_ (Figure 5D) nor the soluble Aβ_42_ cerebral levels (Figure 5E) changed between groups. We also quantified the number of cerebral Aβ deposits by Thioflavin-S staining (Figure 5F): we observed no significant increase of amyloid plaques in the brain of APP/PS1 mice fed a CHOL diet compared to mice fed a STD diet (Figure 5F). In addition, NALFD only decreased by 37% brain *Ide* mRNA expression (Figure 5G), with no change in genes involved in the production, catabolism and clearance of Aβ peptide. Altogether, these results suggest that NAFLD did not significantly impair the production and the clearance of Aβ in the brain of APP/PS1 mice while decreased *Ide* gene expression supports that its catabolism may be slightly reduced.

### 2.6. NAFLD is Associated with Brain Hypoperfusion

N-3 PUFA deficiency, inflammation and oxidative stress have been related to vascular dysfunctions [6,7,28]. Hence, we assessed cerebral perfusion using 7-Tesla MRI. Mice with NAFLD had a significant 20% lower brain perfusion (p < 0.05) than control mice (Figure 6A), in contrast to unchanged perfusion observed when comparing APP/PS1 mice with WT fed with a standard diet (Appendix A). Loss of cerebral microvessels may contribute to brain hypoperfusion. We, therefore, quantified brain microvessel density using collagen IV staining; we observed, however, no difference among groups, neither in the cortex, nor in the hippocampus (Figure 6B).

### 2.7. NAFLD Reduces Cognitive Performance

Finally, we evaluated the impact of NAFLD on cognitive functions of APP/PS1 mice using the Y-maze behavioral paradigm to assess spatial memory. AD mice fed a standard diet exhibit poor performance, showing that 6-month-old APP/PS1 mice are already cognitively impaired, as previously reported [36]. Mice with NAFLD displayed a significant decrease in alternation rate (p < 0.05), indicating a worsened spatial working memory (Figure 7A). Since the number of arm entries in the Y-maze test did not change significantly among experimental groups (Figure 7B), the effects of CHOL diet were not due to exploratory, locomotor, visual, or motivational effects.

## 3. Discussion

The main findings of this study are that a diet-induced NAFLD without obesity in a mouse model of AD (1) imbalances brain cholesterol and fatty acid metabolism, particularly cerebral MUFAs and n-3 PUFAs deficiency, (2) is associated with brain inflammation, senescence and oxidative stress, (3) alters Aβ peptide metabolism only in the liver and the plasma, and (4) reveals cerebral hypoperfusion and a decline in cognitive functions. Although at this stage we cannot conclude on the direct causal influence of NAFLD, these data provide strong evidence that liver failure may accelerate cognitive decline in AD.

The literature has reported that NAFLD was associated with cognitive decline and that liver PUFA (n-3) fatty acid contents strongly correlate with cognitive status in people with AD [31,32], suggesting a link between lipid metabolism in the liver and the functions of the brain. Our hypothesis was that NAFLD could accelerate the progression of AD through disturbed liver, plasma, and brain lipid homeostasis. Achieving this goal is challenging for two reasons: first, the classical approach used to induce liver failure is to work with mice knockout for apolipoprotein E and lipoprotein receptors [37,38]. However, these targets are also expressed in the brain and are involved in lipid homeostasis and in the clearance of Aβ peptide [39,40,41]. Therefore, the use of these transgenic mice is not appropriate to investigate the impact of NAFLD on brain lipid homeostasis and Aβ peptide homeostasis. Second, most of the dietary patterns used to promote NAFLD also induce confounding factors such as obesity and hyperglycemia [42,43]. In this study, we exposed APP/PS1 mice to a diet rich in saturated fatty acids, cholesterol, and cholate that leads to NAFLD without weight gain and plasma glucose level disturbances. Despite being rich in fat, this diet reduces visceral adipose tissue and only modestly decreases insulin sensitivity after long-term feeding [44,45]; our results are therefore in accordance with this phenotype. Since this diet may also induce atherosclerosis in some mouse models [45], we verified that APP/PS1 mice exposed to the diet did not develop atherosclerotic plaques (data not shown). To the best of our knowledge, only one study investigated the potential existence of a liver-brain axis in mice [46]. Although the authors also used a similar diet in their study, they reported a weight gain in APP-transgenic mice [46]. The discrepancy between their results and ours might be related to the use of a hybrid background by backcrossing to C3HeJ×C57BL/6J mice [46] compared to our C57/BL6 mice [33].

Some studies reported that dietary interventions might change lipid contents in the brain [47,48]. This is consistent with our results since more than half of the brain fatty acid levels were significantly different among groups. Interestingly, we observed a deficiency in MUFAs and n-3 PUFAs, including lower DPA and EPA levels, in the brain of mice fed a CHOL diet compared to mice fed a standard diet. Our results were confirmed by a non-targeted lipidomic analysis showing that 31% of lipids significantly different between groups were n-3 and n-6 PUFAs. Since differences were observed in some phospholipids classes, NAFLD may disturb incorporation and hydrolysis of brain membrane fatty acids leading to impaired membrane homeostasis and signaling in the brain [49]. N-3 PUFAs have anti-inflammatory properties and DPA and EPA-enriched diets lower inflammation in the brain [50,51,52,53]. In our mouse model with NAFLD, there was no sign of increased astrogliosis, but there was a higher brain expression of *Tnfα*. Since mRNA levels of the macrophage marker *Cd68* and adhesion molecules *Vcam* and *Icam1* were similar among groups, it is unlikely that macrophage infiltration across the blood brain barrier contributes to *Tnfα* overexpression, confirming that *Tnfα* overexpression in the brain may be related to n-3 PUFAs deficiency [54]. Interestingly, we also found a higher brain mRNA expression of the senescent marker *p21* in mice fed a CHOL diet compared to standard diet. Previous studies reported that *Tnfα* induces *p21* expression in colon cancer cells and hepatocytes [55,56]. Because Tnf-α belongs to the senescence-associated secretory phenotype (SASP) produced by senescent cells, and that through the SASP, senescence begets senescence, the reduction in essential anti-inflammatory n-3 fatty acids associated with NAFLD may initiate (middle-age adults) or exacerbate (older age) the senescence of brain cells that would then synergize to further their accumulation and prematurely exhaust cognitive reserves. We previously reported that a cerebrovascular hemodynamic stress increased brain senescence load [36], while others reported that brain senescent cells contributed to neurodegeneration and cognitive impairment [57]. However, whether low n-3 fatty acids could promote senescence in brain cells susceptible to AD deserves further in vitro investigations.

Studies performed in animal models, old subjects and in AD patients have shown that cerebral perfusion is lower than in healthy subjects [58,59] and that cerebral blood flow decreases with age [60,61]. Herein, we reported that under a standard diet, 6-month old APP/PS1 mice did not shown any alteration in cerebral perfusion. In contrast, lower brain perfusion was revealed in APP/PS1 mice fed a CHOL compared to APP/PS1 mice fed a control diet that supports the critical role of diet-induced NAFLD and lipidomic disturbances. This hypoperfusion was not caused by a reduction in capillaries density, as shown by the lack of change in collagen IV staining among groups. Brain hypoperfusion may also be related to a specific vascular dysfunction: we reported that middle cerebral artery endothelial function is severely impaired in APP/PS1 mice [36] and may promote cerebral artery constriction as suggested by others [61,62]. We quantified the amyloid load in the brain of APP/PS1 mice by thioflavin-S staining and by measuring Aβ_40_ and Aβ_42_ levels in the cerebral soluble fraction. The amount of amyloid deposits or the levels of soluble Aβ_40_ and Aβ_42_ peptides were similar among groups despite a decrease in brain mRNA expression of *Ide* (that degrades Aβ peptide), and lower plasma Aβ_40_ and Aβ_42_ levels. Therefore, our results do not support the hypothesis that a change in brain Aβ peptide is responsible for NAFLD-dependent microvascular dysfunction and cognitive impairments in AD [63,64], further supporting the impact of lipid metabolism on brain function and thus cerebral perfusion. Clearly, however, chronic amyloid pathology in the brain increases stress sensitivity by reducing cerebrovascular and cognitive reserves [33,36].

Other mechanisms than Aβ alterations seem, indeed, to be involved in NAFLD-dependent cerebral hypoperfusion observed in APP/PS1 mice. N-3 PUFAs and MUFAs improve endothelial function [65,66] and n-3 PUFAs ameliorate brain perfusion in patients with mild cognitive impairments [28]. Therefore, their reduction in the brain could promote cerebral hypoperfusion in mice with NAFLD. Another argument in favor of the involvement of FAs in cerebrovascular dysfunction is the higher *Cox-2* mRNA expression observed in the brain of APP/PS1 mice fed a CHOL diet. Cox-2 is a cyclooxygenase overexpressed in the brain of AD patients [67] and in senescent cells [68] that metabolizes arachidonic acid to form prostaglandin H_2_ then converted into thromboxane TxA_2_, a potent vasoconstrictor [69]. Moreover, its metabolites are also involved in the control of the cerebral blood flow [70]. Even if our data do not support a direct link between *Cox-2* overexpression and cerebral hypoperfusion, our results reinforce the idea that brain PUFAs and particularly n-3 may be the mediators linking NAFLD and cerebral hemodynamics. Interestingly, EPA decreases Cox-2 expression [71] that is upregulated following inflammation [72]. In our model, brain EPA levels were lower in APP/PS1 mice with NAFLD, and this could explain the overexpression of *Cox-2*. Our results combining higher *Cox-2* and higher *Nox-2* mRNA expression in the brain are in accordance with studies showing that both are important sources of oxidative stress [73,74]. One study performed in the Tg2576 amyloid mouse model showed that Nox-2 deletion suppressed oxidative stress, cerebrovascular dysfunction and cognitive deficits, independently of brain Aβ peptide levels or amyloid plaques [74]. Therefore, in the present study, the diet-induced *Cox-2* and *Nox-2* overexpression may contribute to the cerebral hypoperfusion observed in APP/PS1 mice with NAFLD.

Finally, we assessed memory performance of APP/PS1 mice using the Y-Maze. Our results are in accordance with our previous study where the alternation rate of 6-month old APP/PS1 mice was about 65% versus 78% in wild type mice [36]. Therefore, even if 6-month old APP/PS1 mice already expressed cognitive impairment, the Y-maze test reveals in APP/PS1 mice with NAFLD even lower spontaneous alternations (Figure 7A). This premise may suggest that NAFLD, through a negative impact on spatial memory, may contribute to the exacerbation of cognitive alterations. Our observations are reinforced by many studies showing memory impairments following high-fat diet consumption [75,76,77]. It is the first study, however, to characterize simultaneously blood brain perfusion, inflammation, senescence, oxidative stress, amyloid pathology and lipid homeostasis in the peripheral and central compartments and to show that liver failure alone is sufficient to impair brain health, independently of cerebral amyloid pathology.

To investigate in vivo the specific links existing between the liver and the brain is challenging and our study has some limitations. Even if we could not eliminate all confounding factors, our model permitted nevertheless to isolate the hepatic alterations by preventing hyperglycemia, obesity, and the increase in adipose tissue mass, which is the primarily production site of pro-inflammatory cytokines. A limitation of our study is that molecular analyses were performed on whole brain homogenates and not on specific brain structures, microvessels, or parenchyma. Moreover, experimenters could not be blinded to the dietary status during behavioral tests and MRI analyses due to the difference in food color. Finally, it is unclear which parameter (i.e., brain lipid homeostasis, senescence, inflammation, oxidative stress) is responsible for brain hypoperfusion and cognitive decline, but likely all parameters synergize to contribute to brain damage. Further investigations are needed to characterize the underlying mechanisms linking lipid metabolism in the liver and the functions of the brain.

In conclusion, we demonstrate that the brain of APP/PS1 mice is vulnerable to metabolic disorders induced by NAFLD. Importantly, we report in APP/PS1 mice that NAFLD reduces plasma, liver, and brain n-3 PUFAs contents, while MUFAs were only decreased in the brain. These observations were related to an increase of numerous hallmarks of AD such as inflammation, senescence, oxidative stress, and vascular dysfunction, leading to worsened cognitive impairments, regardless of amyloid pathology in the brain. Although the brain pathology developed in this mouse model does not entirely recapitulated the AD disease, our study supports the concept of a liver-brain axis and suggests that dysregulated plasma and brain fatty acid metabolism associated with NAFLD contributes to the progression of brain disease in mice that have impaired cerebrovascular and cognitive reserves.

## 4. Material and Methods

### 4.1. Animal Experiments

Six-weeks-old APP/PS1 male mice from our colony were used [36]. All animals were housed individually in a pathogen free animal facility with 12-h light/dark cycles. A standard (STD) diet containing 4%, 18% and 64% (w/w) of fat, protein, and carbohydrates, respectively (D12337; Research Diet, New Brunswick, US) and a Paigen diet containing 16%, 21% and 46% (w/w) of fat, protein, and carbohydrates, respectively, and supplemented with both cholesterol (1.25% w/w) and cholate (0.5% w/w) (CHOL) (D12336; Research Diet, New Brunswick, US) were given *ad libitum* for 18 weeks. Mice were randomly assigned in STD diet group (body weight = 20.9 ± 0.3 g) (n = 42) or CHOL diet group (20.8 ± 0.3 g) (n = 40). The diets induced no side effect. Body weight and food consumption were measured weekly. Mice were killed at the age of six months by exsanguination under anesthesia (100 mg/kg ketamine and 10 mg/kg xylazine i.p.) following 15 h of fasting. The liver and brain were snap-frozen, and plasma collected and kept at −80 °C until further analysis. All animal experiments were performed in accordance with the “Guide for the Care and Use of Experimental Animals of the Canadian Council on Animal Care” and were approved by the Montreal Heart Institute Ethics Committee (identification code: 2015-62-01). Results were reported according to the ARRIVE (Animal Research: Reporting of In Vivo Experiments) guidelines to the best of our ability.

### 4.2. Lipids Analyses

#### 4.2.1. Untargeted Lipidomics Analysis Using LC-MS

Lipid extraction and analysis were performed as previously described with slight modifications [78]. Briefly, lipids were extracted from liver (20 mg) or brain (40 mg) tissues, which had been spiked with six internal standards. Samples were injected onto a 1290 Infinity HPLC coupled with a 6530 (liver tissues) or 6550 (brain tissues) accurate mass QTOF (Agilent, Santa Clara, CA, USA) equipped with a dual electrospray ion source.

Lipids were eluted on a Zorbax Eclipse plus C18, 2.1 mm × 100 mm, 1.8 µm (Agilent, Santa Clara, CA, USA) kept at 4 °C with a gradient of 83 min. Liver tissues were analyzed in positive scan mode while brain tissues were analyzed in both negative and positive scan mode. Each feature characterized by a specific mass and retention time, were extracted using Mass Hunter B.07.00 software (Agilent, Santa Clara, USA). The data processing consisted of (1) application of a frequency filter of 80% on feature’s presence in one condition, (2) normalization of signal intensities using cyclic loess algorithm, (3) imputation of missing values using KNN5 and (4) correction of batch effect using Combat according to animal’s sacrifice day.

Data were analyzed with Mass Profiler Professional (MPP; Agilent Technologies, Santa Clara, CA, USA) using unpaired student’s t-test followed by Benjamini Hochberg correction. Mass spectrometry signals that discriminated the two conditions based on a corrected p-value of 0.375 (FDR < 37.5%) and a fold change > 1.2, were subjected to tandem mass spectrometry for lipid identification.

Cholesterol, hydroxycholesterol, desmosterol, and lathosterol, were identified separately and validated using their corresponding standards.

#### 4.2.2. Targeted Fatty Acids Analysis Using GC-MS

Liver and brain tissues were processed for quantitative profiling of fatty acids by gas chromatography-mass spectrometry (GC-MS) using a previously described method [79]. In brief, pulverized tissues (25 mg) were incubated overnight at 4 °C in a solution of chloroform/methanol (2:1) containing 0.004% butylated hydroxytoluene (BHT), filtered through gauze, dried under nitrogen gas and re-suspended in hexane/chloroform/methanol (95:3:2). Triglycerides and phospholipids were separated on the aminoisopropyl column (Agilent, Santa Clara, CA, USA) as previously described in detail [80]. All samples were dried under nitrogen gas after the addition of internal standards and were re-suspended in hexane/methanol (1:4) containing 0.004% BHT.

Fatty acids were analyzed as their methyl esters after a direct trans-esterification with acetyl chloride/methanol on a 7890B gas chromatograph coupled to a 5977A Mass Selective Detector (Agilent Technologies, Santa Clara, CA, USA) equipped with a capillary column (J&W Select FAME CP7420; 100 m × 250 µm inner diameter; Agilent Technologies, Santa Clara, CA, USA) and operated in the positive chemical ionization mode using ammonia as the reagent gas. Samples were analyzed under the following conditions: injection at 270 °C in a split mode using high-purity helium as the carrier gas at the following temperature gradient: 190 °C for 25 min, increased by 1.5 °C/min up to 236 °C. Fatty acids were analyzed as their [M+NH_3_]^+^ ion by selective ion monitoring and fatty acids concentrations were calculated using standard curves and isotope-labeled internal standards.

### 4.3. Real-Time Quantitative Polymerase Chain Reaction

Total ribonucleic acid RNA was extracted from liver and brain using the RNeasy lipid tissue mini-kit as specified (Qiagen, Toronto, ON, Canada). Reverse-transcription was performed using the M-MLV reverse transcriptase kit (Thermo Fisher Scientific, Waltham, MA, USA). Quantitative polymerase chain reaction (qPCR) was performed using the EvaGreen qPCR Mastermix (Mastermix-LR; Thermo Fisher Scientific, Waltham, MA, USA). The delta CT (threshold cycles) method (ct value is defined as the number of amplification cycles required to reach a fixed signal threshold) was used for analysis of relative gene expression using *cyclophilin A* as the housekeeping gene. Samples with CT values > 35 were excluded from the analyses to avoid unreliable results.

### 4.4. Biochemical Analysis

Plasma from APP/PS1 mice was obtained by centrifugation (12,000 rpm for 15 min) and stored at –80 °C. Plasma cholesterol and glucose were measured at the Clinical Biochemistry Laboratory of the Montreal Heart Institute (Montreal, QC, Canada) and plasma Aβ_40_ (KHB3481; Thermo Fisher Scientific, Waltham, MA, USA) and Aβ_42_ (KHB3544; Thermo Fisher Scientific, Waltham, MA, USA) levels were quantified using ELISA tests, according to the manufacturer’s instructions.

### 4.5. Magnetic Resonance Imaging Methods

MRI was performed on a 30 cm 7T horizontal MR scanner (Agilent, Palo Alto, CA, USA) with the mouse in prone position, with a 12 cm inner diameter gradient coil insert, gradient strength 600 mT/m, rise-time 130 ms, as previously described [36]. Briefly, a 2-channel receive-only surface coil positioned over the mouse brain was used in combination with a quadrature transmit/receive birdcage coil with an internal diameter of 69 mm (RAPID Biomedical, Rimpar, Germany). Anesthesia was maintained with 1.4–2.2% isoflurane in 30% oxygen in air and body temperature was maintained at 37.0 °C using a warm air fan (SA Instruments, Stony Brook, NY, USA). Respiration (target = 100, allowed range before adjusting isoflurane = 80–120 respiration-per-minute) and heart rate were monitored, the latter with a pulse-oximeter.

An anatomical image was acquired with a 3D true free induction with steady-state precession (TFISP) sequence [81] at 100 µm isotropic resolution (TR = 5.0 ms/TE = 2.5 ms, 16 frequencies, 22 min scan time), used for co-registration. Then a 3D amplitude-modulated continuous arterial spin labeling scan (amCASL) was run, with TR = 3.0 s, 1.0 s labeling duration, 60 × 54 × 48 matrix, 300 × 333 × 333 µm resolution, 23-min scan time. The tagging plane was kept in a fixed position nominally perpendicular to the carotids.

### 4.6. MRI Data Analysis

Perfusion was calculated voxel-by-voxel using Formula 1 in [82], with some parameters assumed constant: brain/blood partition coefficient = 0.9 mL/g, mouse arterial blood transit time = 0.08 s, tagging efficacy = 0.67, T_1b_ = 2.3 s, T_1_ = 1.53 s, T_1sat_ = 0.57 s, M_a_^z^(w) = 0.48 s.

The anatomical scans were first manually co-registered rigidly using ITK-SNAP [83] to a previously generated mouse template, then co-registered non-linearly with advanced normalization tools (ANTs) [84]. This provided the ANTs transformations to co-register the perfusion images. Co-registration results were inspected in details, and registration parameters were stored as potential confounding variables.

Two-sample t-test statistical parameter mapping (SPM) voxel-by-voxel analyses were run on the perfusion images. The observation of weak effects distributed over the whole brain suggested a region-of-interest (ROI) study, averaging over the whole brain. One ROI was chosen to reflect the whole brain, including the cerebellum, but not the brain stem, the olfactory bulb or the ventricles. The MarsBar toolbox in SPM was used for the ROI statistics and to extract the average perfusion data on the ROI. Some potential confounds were included as regressors of no interest (4: one rotation, one translation and one dilation from the co-registration parameters, and the transmit B1 field strength). A confidence of 95% to reject null hypotheses was considered significant.

### 4.7. Y-Maze

Spatial working memory was assessed in a Y-maze, by recording spontaneous alternations behavior. The maze was made of opaque Plexiglas, and each arm was 40-cm long, 16-cm high, and 9-cm wide, and positioned at equal angles. Mice were placed at the end of one arm and allowed to freely explore the three arms for 5 min. The series of arm entries were recorded visually, and arm entry was validated when the base of the tail of the mouse was completely placed in the arm. Alternation score was calculated as the ratio of the number of triads to the total number of possible alternations (defined as the number of arm entries minus two) [85].

### 4.8. Amyloid Deposits Staining

Brain cryosections were fixed in 4% (w/v) paraformaldehyde (PFA) in phosphate-buffered saline (PBS) for 15 min at room temperature. After washing in PBS, the sections were incubated 15 min at room temperature with 0.2% (w/v) thioflavin-S (Sigma-Aldrich, Saint-Louis, MO, USA) solution in 80% ethanol followed by washing in 80% ethanol (1 min), 70% ethanol (1 min). Slices were washed twice in distilled water, covered by mounting medium and coverslips for imaging. All quantifications were performed using Image J software, version 1.8.

### 4.9. Cerebral Microvessel Density Staining

After brain fixation in 4% PFA, brain sections were incubated for 1 h at room temperature in PBS containing 0.5% (v/v) Tween-20 and 4% (w/v) bovine serum albumin (BSA). After washing with PBS, slides were incubated at 4 °C for 24h in PBS containing the polyclonal rabbit anti-collagen IV antibody, (1:100, Abcam ab19808, Cambridge, UK) with 1% BSA. After washing with PBS, brain slices were incubated at room temperature for 2 h in PBS containing Alexa Fluor 647 donkey anti-rabbit IgG (1:500, Abcam, Cambridge, UK) and DAPI solution (1:600, Thermo Fisher Scientific, Waltham, MA, USA) with 1% BSA. After washing, slides were covered by mounting medium and coverslips for imaging. Cerebral microvessel density was quantified using Image J software version 1.8 as a percentage of collagen IV-positive area per ROI as previously [36].

### 4.10. Statistical Analyses

The minimal number of mice required per experiment was based on the variability of the data collected during our previous studies [36]. Student’s t-tests were performed with Prism 6 (La Jolla, CA, USA) to assess differences between groups. Data are expressed as means ± SD. Statistical significance was set at p < 0.05. The interquartile range (IQR) was calculated as the third quartile (Q3) minus the first quartile (Q1). Values below Q1−1.5 IQR or above Q3+1.5 IQR were excluded from the analyses.

## Figures and Tables

**Figure 1 metabolites-09-00104-f001:**
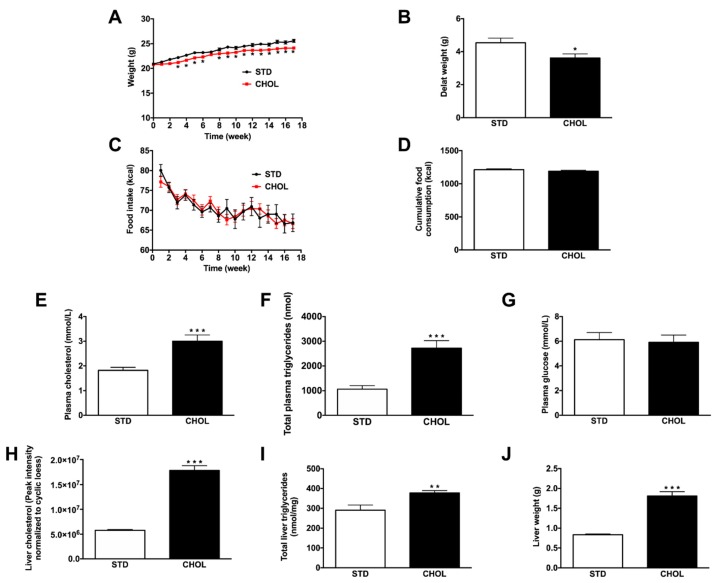
Fatty acid and cholesterol-enriched diet induces NAFLD without obesity in APP/PS1 mice. (**A**) Weight, (**B**) delta weight gain, (**C**) food intake, (**D**) cumulative food consumption (n = 40–42 animals per group), (**E**) plasma cholesterol, (**F**) triglycerides, (**G**) glucose levels (n = 9–10 animals per group) and (**H**) liver cholesterol, (**I**) triglycerides levels (n = 10 animals per group) and (**J**) liver weight (n = 10–11 animals per group) of APP/PS1 mice fed a standard diet (STD) or a cholesterol-enriched diet (CHOL). Results are means ± SD. Unpaired t-tests were performed to assess differences between groups; * p < 0.05, ** p < 0.01, *** p < 0.001.

**Figure 2 metabolites-09-00104-f002:**
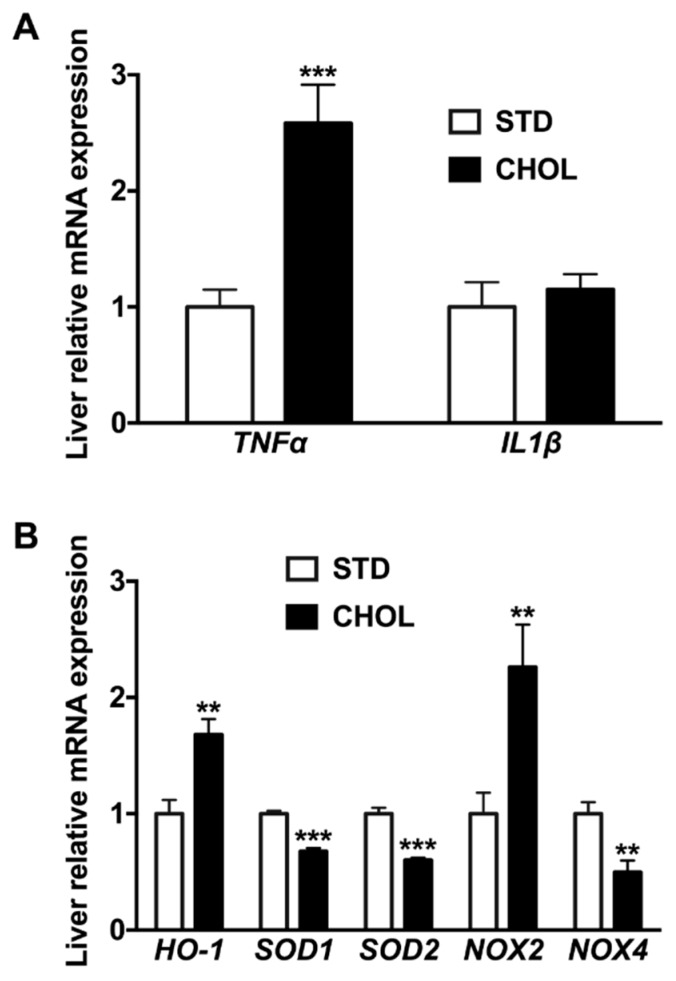
Fatty acid and cholesterol-enriched diet induces inflammation and oxidative stress in the liver of APP/PS1 mice. Gene expression of (**A**) inflammation and (**B**) oxidative stress markers in the liver from APP/PS1 mice fed a standard diet (STD) or a cholesterol-enriched diet (CHOL). Results are means ± SD. Unpaired t-tests were performed to assess differences between groups; (n = 10 animals per group); ** p < 0.01, *** p < 0.001.

**Figure 3 metabolites-09-00104-f003:**
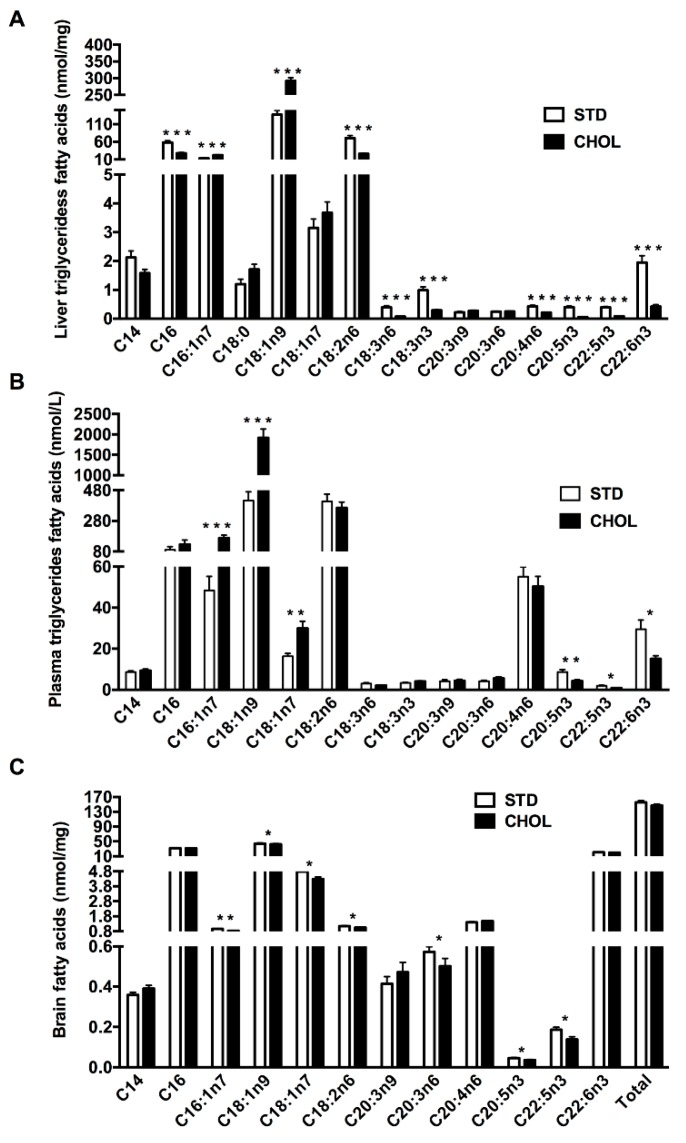
NAFLD imbalances plasma, liver and brain lipid metabolism in APP/PS1 mice. (**A**) Liver and (**B**) plasma fatty acid profiles in triglycerides fractions and (**C**) brain fatty acid profiles in APP/PS1 mice fed a standard diet (STD) or a cholesterol-enriched diet (CHOL). Results are means ± SD. Unpaired t-tests were performed to assess differences between groups; (n = 10 animals per group). All animals were included in analyses. * p < 0.05, ** p < 0.01, *** p < 0.001.

**Figure 4 metabolites-09-00104-f004:**
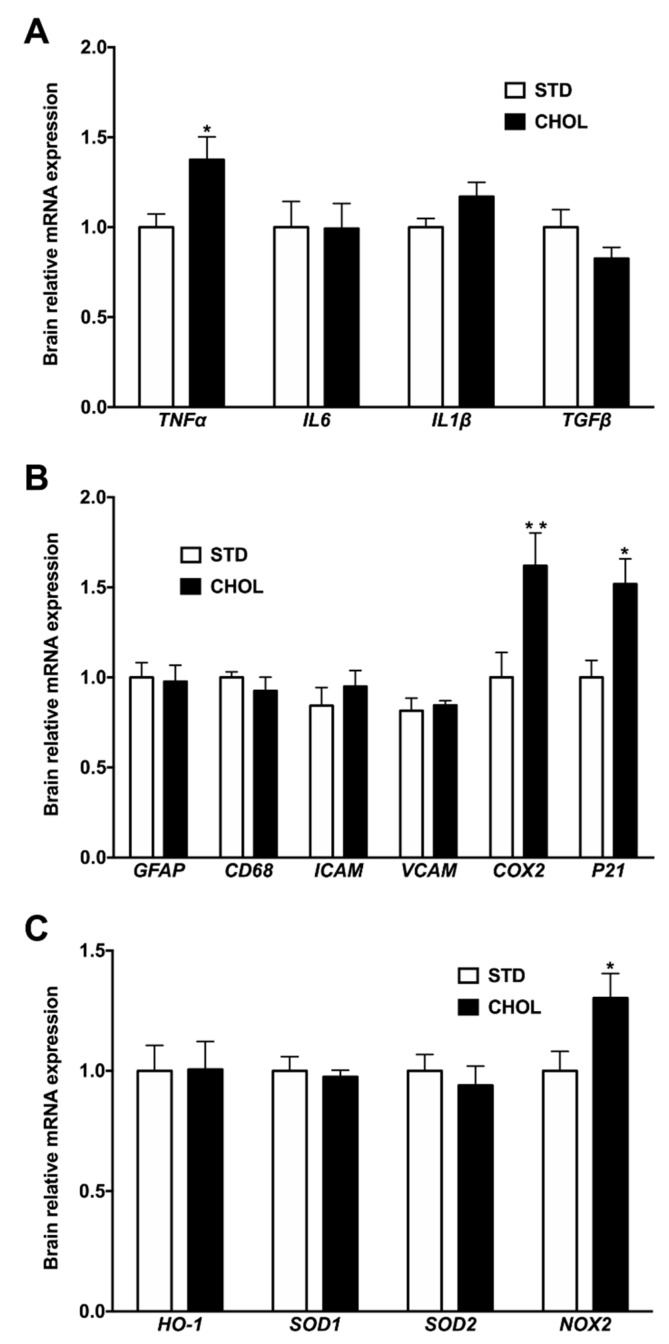
NAFLD is associated with brain inflammation, senescence, and oxidative stress. Gene expression of inflammation (**A**), neuro-inflammation and senescence (**B**) and oxidative stress markers (**C**) in the brain from APP/PS1 mice fed a standard diet (STD) or a cholesterol-enriched diet (CHOL). Results are means ± SD. Unpaired t-tests were performed to assess differences between groups; (n = 9 animals per group); * p < 0.05, ** p < 0.01, *** p < 0.001.

**Figure 5 metabolites-09-00104-f005:**
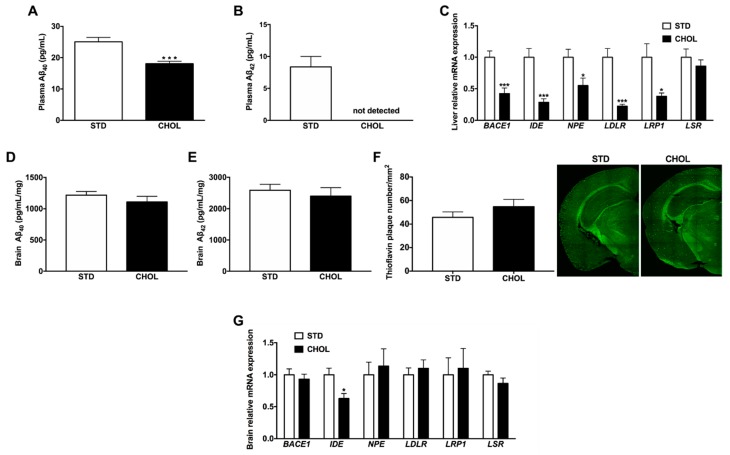
NAFLD dysregulates amyloid beta peptide metabolism in the liver, the plasma but not in the brain of APP/PS1 mice. (**A**) Plasma Aβ_40_, (**B**) plasma Aβ_42_ (n = 13–15) and (**C**) qPCR analysis of genes involved in Aβ metabolism in the liver (n = 10 per group), (**D**) brain Aβ_40_ and (**E**) brain Aβ_42_ levels (n = 8 per group), (**F**) staining with thioflavin-S for the detection of amyloid deposits in coronal brain sections (n = 7 per group) and (**G**) qPCR analysis of genes involved in Aβ metabolism in the brain (n = 9 *per* group) from APP/PS1 mice fed a standard diet (STD) or a cholesterol-enriched diet (CHOL). Results are mean ± SD. Unpaired t-tests were performed to assess differences between groups; * p < 0.05, ** p < 0.01, *** p < 0.001.

**Figure 6 metabolites-09-00104-f006:**
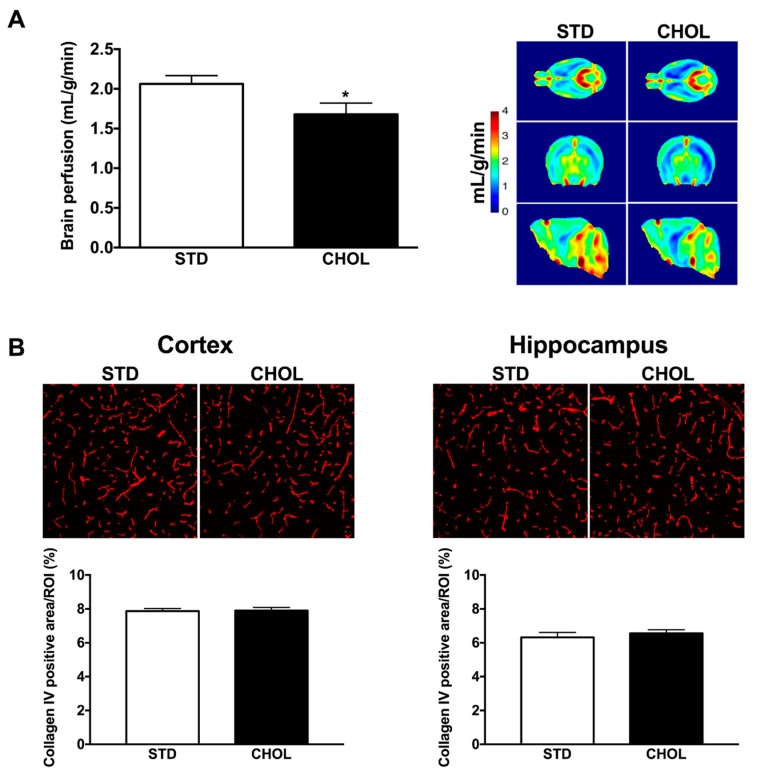
NAFLD induces brain hypoperfusion in APP/PS1 mice. (**A**) Quantification of brain perfusion by 7-T IRM in APP/PS1 mice fed a standard diet (STD) (n = 16) or fed a cholesterol-enriched diet (CHOL) (n = 13). Typical 7-IRM images are shown. (**B**) Brain capillary density (% of positive staining per region of interest, ROI) was quantified by immunohistochemistry staining of collagen IV in the cortex and hippocampal areas of APP/PS1 mice fed a STD diet (n = 7) or a CHOL diet (n = 7);. Results are mean ± SD. Unpaired t-tests were performed to assess differences between groups; * p < 0.05.

**Figure 7 metabolites-09-00104-f007:**
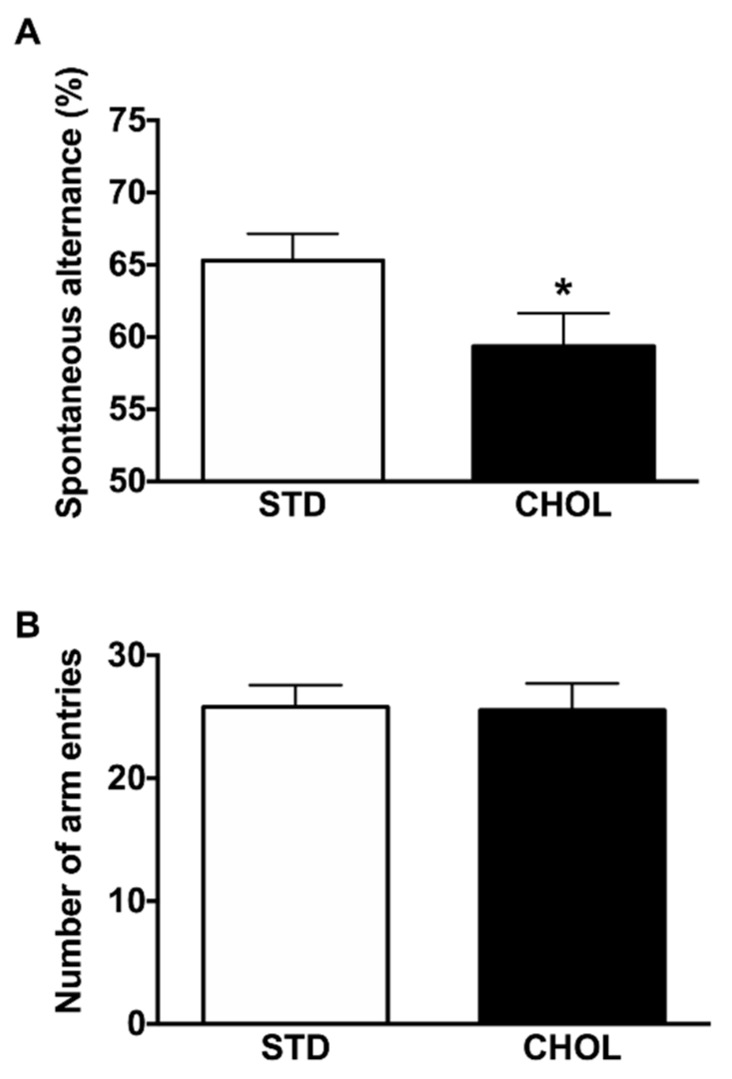
NAFLD affects spatial working memory in APP/PS1 mice. (**A**) Evaluation of spatial working memory and (**B**) exploratory activity with the Y-maze test in APP/PS1 mice fed a standard diet (STD) (n = 15) or a cholesterol-enriched diet (CHOL) (n = 13). Results are means ± SD. One-tailed unpaired t-tests were performed to assess differences between groups; * p < 0.05.

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
