# Peer review of "Non-Alcoholic Fatty Liver Disease, and the Underlying Altered Fatty Acid Metabolism, Reveals Brain Hypoperfusion and Contributes to the Cognitive Decline in APP/PS1 Mice"

_metabolites, 2019, doi:10.3390/metabo9050104_

Round 1

Reviewer 1 Report

Metabolites submission: 505921

Nonalcoholic fatty liver disease (NAFLD) is most common cause of chronic liver disease which also contributes to cognitive impairment for which underlying mechanisms are not yet fully understood. Current submission ‘Non-alcoholic fatty liver disease, and the underlying disturbed lipid metabolism, reveals brain hypoperfusion in APP/PS1 mice’ by Anthony Pincon et al, is definitely interesting and presents expansion of research into NAFLD and Alzheimer’s disease (AD).

Authors hypothesized that diet induced NAFLD condition should manifest as cerebral hypo perfusion along with changes in brain and liver lipid metabolism. Towards testing this hypothesis, they chose amyloid precursor protein/presenilin-1 (APP/PS1) transgenic mice model for Alzheimer’s disease (AD) and fed with standard or saturated fatty acid, cholesterol and cholate (CHOL) diet for six months to induce NAFLD. Subsequently, they analyzed plasma, brain and liver tissue samples to identify fatty acid content and composition, gene expression levels and Ab peptide metabolism. Liver and brain tissue samples show that mice fed on CHOL diet show decreased levels of n-PUFAs, and increased levels of MUFA. At the same time, they also observed increased expression of genes indicative of brain inflammation and stress. As hypothesized, NAFLD induction shows reduced production, catabolism and clearance of hepatic and plasma Ab peptide where as in brain tissues, it is not significantly impaired. They also find that NAFLD induces cerebral hypoperfusion. Furthermore, Y-maze tests confirm NAFLD aggravates cognitive decline in APP/PS1 mice.

Overall, the manuscript is well written. Experiments are well designed and conducted. Inferred results are appropriate and discussion is brief and up to point. Literature citation is comprehensive and covers important citations. Figures, legends and tables explain the results pretty well and are consistent with the text. Therefore, I feel this manuscript will be of good interest to the researchers in the community.

Minor points to improve quality of manuscript:

·         In-consistency in fonts (lines 221-229) vary from rest of text

·         AD in abstract should be abbreviated in first mention

·         N-3 PUFA should be corrected as n-3 PUFA (line 308)

·         It should be section 3.1 instead of 4.1 (line 358)

·         References:  mix of headline style and sentence style capitalization was used. Should make them all in one style

Author Response

We are grateful to the Reviewer for the careful evaluation of our manuscript  and the positive comments.

We therefore improved the quality of the manuscript as requested by the Reviewer:

- In-consistency in font was corrected

- In the abstract, the abbreviation AD is now present

- Line 308 N-3 has been corrected by n-3

- The references are now uniformed

Reviewer 2 Report

The authors present a very interesting and comprehensive manuscript entitled “Non-alcoholic fatty liver disease, and the underlying disturbed lipid metabolism, reveals brain hypoperfusion in APP/PS1 mice”.  In this study they report the use of multiple techniques to characterize blood brain perfusion, inflammation, senescence, oxidative stress, amyloid pathology and lipid homeostasis in the peripheral and central compartments.  I found the manuscript to be very well written with some typos throughout, which I am assuming with be caught by the editorial team.

As mentioned, I found the manuscript to be very comprehensive in terms of methodologies employed and insightful.  However, I do have some questions for the authors.

Results:

Figures 7’s caption.  Surely this cannot be correct based on Figure 7B?

Discussion:

Lines 236-241:  Can the authors comment on how much of this causative or correlative? Discuss in regard to points 1-4.

Lines 327-330:  Are the authors accurate here?  Based again on Figure 7B, this would seem to be incorrect.

Line 249: “highly vulnerable” is an overstatement.

Lines 349-356: Not to diminish the study in any way (as mouse models have been very useful for studying AD), I would suggest reframing this paragraph to emphasize that you have proved as such in mouse brain; but these are mice and do not develop AD but simply pathology similar to that observed in PM human brain.

Materials and Methods:

In section 3.2 maybe change the shortened titles of LC-MS analysis and GC-MS analysis to include what you are analyzing?  There are different types of lipids after all. GC-MS analysis of FAME’s? Not sure if they were methylated form the description-merely a suggestion.

You discuss a power analysis, but no information is provided on said analyses.  Was the study sufficiently powered?

Author Response

We are grateful to the Reviewer for the careful evaluation of our manuscript  and the positive comments.

1. Figure caption for figure 7 and lines 327-330

Response: we agree that the old title was an overstatement. However, as regards of the Figure 7A, we cannot exclude some cognitive alterations. Therefore, we modulate our title that is now: "NAFLD affects spatial working memory in APP/PS1 mice".

In line with changes in the latter title, we modulate our conclusion lines 327-330 that we finally judged overstated: "the Y-maze test reveals in APP/PS1 mice with NAFLD low spontaneous alternations (Fig. 7A). This premise may suggest that NAFLD, through a negative impact on spatial memory, may contribute to the exacerbation of cognitive alterations".

2.Line 236-241. Is it causative or associative?

The mouse model used, the APP/PS1 mice, are known to develop cognitive alterations in absence of any confounding factor. Is it, thereby difficult to conclude on the causative impact of NAFLD on the development of AD. However, given that NAFLD unmask brain hypoperfusion in APP/PS1 mice, we can strongly suggest that NAFLD accelerate the cognitive decline. We therefore completed the first paragraph of the discussion as follows: "Although at this stage we cannot conclude on the direct causal influence of NAFLD, these data provide strong evidence that liver failure may accelerate cognitive decline in AD".

3. line 349, highly vulnerable is an overstatement

We removed highly.

4. Lines 349-356, emphasize that our findings were observed in mouse

We totally agreed that  this mouse model does not fully develop AD. We therefore add some precision in the conclusion as follows: " we demonstrate that the brain of APP/PS1 mice is vulnerable..." and "Although the brain pathology developed in this mouse model does not entirely recapitulated the AD disease, our study supports the concept of a liver-brain axis and suggests that dysregulated plasma and brain fatty acid metabolism associated with NAFLD contributes to the progression of brain disease in mice that have impaired cerebrovascular and cognitive reserves".

5. Section 3.2. changes title related to the mass spectrometry-based analyses

The new title of section 3.2.1. is now "Untargeted lipidomics analysis using LC-MS"

The new title of section 3.2.2. is now "Targeted fatty acids analysis using GC-MS"

6. Details on power analysis?

Actually, this was a mistake and was corrected as follows: The minimal number of mice required per experiment was based on the variability of the data collected during our previous studies [36]